# Cut-Off Values in the Prediction of Success in Olympic Distance Triathlon

**DOI:** 10.3390/ijerph17249491

**Published:** 2020-12-18

**Authors:** André Bonadias Gadelha, Caio Victor Sousa, Marcelo Magalhaes Sales, Thiago dos Santos Rosa, Marti Flothmann, Lucas Pinheiro Barbosa, Samuel da Silva Aguiar, Rafael Reis Olher, Elias Villiger, Pantelis Theodoros Nikolaidis, Thomas Rosemann, Lee Hill, Beat Knechtle

**Affiliations:** 1Seção de Educação Física (SEF), Colégio Militar de Brasília (CMB), Brasília, DF 70790-020, Brazil; andrebonadias@gmail.com; 2Graduate Program in Physical Education, Catholic University of Brasília, Brasília, DF 71966-700, Brazil; thiagoacsdkp@yahoo.com.br (T.d.S.R.); lduarte.barbosa@gmail.com (L.P.B.); ssaguiar0@gmail.com (S.d.S.A.); 3College of Arts, Media & Design, Bouvé College of Health Sciences, Northeastern University, Boston, MA 02115, USA; cvsousa89@gmail.com; 4Physical Education Department, Goiás State University, Quirinópolis, GO 75860-000, Brazil; marcelomagalhaessales@gmail.com; 5Graduate Program in Human Movement and Rehabilitation of the University Center of Anápolis—UniEvangélica, Anápolis, GO 75083-515, Brazil; 6Miller School of Medicine, University of Miami, Miami, FL 33146, USA; martiflothmann@med.miami.edu; 7Physical Education Department, Centro Universitário do Planalto Central Apparecido dos Santos, Gama, DF 72445-020, Brazil; rflolher@gmail.com; 8Institute of Primary Care, University of Zurich, 8091 Zurich, Switzerland; evilliger@gmail.com (E.V.); thomas.rosemann@usz.ch (T.R.); 9School of Health and Caring Sciences, University of West Attica, 12243 Athens, Greece; pademil@hotmail.com; 10Department of Gastroenterology and Nutrition, Department of Pediatrics, McMaster University, Hamilton, ON L8N 3Z5, Canada; hilll14@mcmaster.ca; 11Medbase St. Gallen Am Vadianplatz, 9000 St. Gallen, Switzerland

**Keywords:** endurance, swimming, cycling, running, triathlon, Olympic distance

## Abstract

Cut-off points and performance-related tools are needed for the development of the Olympic distance triathlon. The purposes of the present study were (i) to determine cut-off values to reach the top three positions in an Olympic distance triathlon; (ii) to identify which discipline present the highest influence on overall race performance and if it has changed over the decades. Data from 1989 to 2019 (*n* = 52,027) from all who have competed in an official Olympic distance triathlon events (World Triathlon Series and Olympics) were included. The cut-off value to achieve a top three position was calculated. Linear regressions were applied for performance trends overall and for the top three positions of each race. Men had cut-off values of: swimming = 19.5 min; cycling = 60.7 min; running = 34.1 min. Women’s cut-off values were: swimming = 20.7 min; cycling = 71.6 min; running = 38.1 min. The running split seemed to be the most influential in overall race time regardless of rank position or sex. In conclusion, cut-offs were established, which can increase the chances of achieving a successful rank position in an Olympic triathlon. Cycling is the discipline with the least influence on overall performance for both men and women in the Olympic distance triathlon. This influence pattern has not changed in the last three decades.

## 1. Introduction

Olympic distance triathlon (1.5 km/0.93 mile swim; 40 km/24.9 mile cycle; 10 km/6.2 mile run) has become very popular since it debuted at Sydney’s Olympic Games in 2000 [1]. Its roots date back to 1989 when the International Triathlon Union (ITU) changed the single race Triathlon World Cup to a World Triathlon Series (WTS) annual circuit in 2009 [2,3]. Briefly, from 1989 to 2008, the world championship was a set by a single race. Thus, in 2009, the WTS was created and World Championship was based on several triathlon races and a great final, demanding more commitment from athletes to reach sport success. Each discipline may require specific training goals in every triathlon race to achieve the best possible performance aligned with a pre-established training strategy and in-race tactics [3].

Despite the importance of good performance in all three disciplines (i.e., swimming, cycling, and running, separately) for success in a triathlon race, it has been reported that the cycling split is the discipline that has the most significant influence on overall race time in an Ironman triathlon (3.8 km/2.4 mile swim; 180 km/111.9 mile cycle; 42 km/26.1 mile run) [2,4], however, this may not be the case in an Olympic distance triathlon—previous reports have shown that cycling is the discipline which possibly has the lowest influence on overall race time [2,4]. Thus, for both training schedules and competitions, the strategies for each triathlon distance should be vastly different from each other [5]. For example, the draft-allowed cycle split leads to the formation of cycling pelotons, so a faster swim may allow the athlete to cycle in the leading (and often faster) pack [6]. Utilizing the draft or slipstream created by the cycling peloton allows the athlete to travel faster while saving energy for the running split [3,4,7]. On the other hand, nondraft cycling requires a self-imposed pacing strategy within all three disciplines, not depending on a faster swim to be a faster cyclist [8,9]. Indeed, recent research showed that swimming had the lowest influence on overall race time among the fastest (sub-8-hour performances) Ironman athletes [10].

Until the 1990s, drafting was not allowed in triathlon world championships, thus cut-off points may be different from drafting races, so the evolution of the sport must be taken into account in proposals for cut-offs. Identifying cut-off points of which individuals would be at greater chance for a podium finish in triathlon, specifically for an Olympic distance, would be an important factor for athletes’ planning, training schedules and competition strategies [11]. The identification of cut-off values in the prediction of a podium finish for each triathlon discipline would be helpful for the earlier and more accurate adjustments in the planning of each athlete, permitting implementation of specific strategies during the training schedule [12]. Currently, there are no studies that have identified cut-off points to achieve a top three position in a triathlon race. However, this information can be useful, since cut-offs of split race times can help athletes and coaches to establish racing strategies and training goals for an optimal race pace.

Although each race may present unique environmental characteristics [7], a general idea of the desired race pace would greatly benefit emerging professional triathletes and their coaches [13,14]. Additionally, the understanding of which split discipline is most important can add important information for this triathlon distance [14,15,16]. To the best of our knowledge, no studies provided the aforementioned information for Olympic distance triathlon. Noteworthy, these analyses can provide support for further scientific investigations, as well bring insights that may be useful supplements for coaches that are engaged in training athletes for and Olympic distance triathlon. Therefore, the aims of the present study were: (i) to determine cut-off values to reach the top three positions in an Olympic distance triathlon; (ii) to identify which discipline presents the highest influence on overall race performance and if it has changed over the decades.

## 2. Materials and Methods

### 2.1. Ethics Approval

This study was conducted according to the international standards and as described in [17]. This study was also approved by the local institutional review board (01/06/2010) with a waiver of the requirement for informed consent of the participants as the study involved the analysis of publicly available data.

### 2.2. Sample

All data were sourced from the official publicly available results of the World Triathlon Series (WTS) events from 1989 to 2019 [18] and triathlon races held during the Olympic Games (2004 to 2016) [19]. In order to obtain the results efficiently, in a standardized fashion and with minimized human error, a custom python script was written and used to download data from the desired events. The standard distance for these events was a 1.5 km/0.93 mile swim, 40 km/24.9 mile cycle, and 10 km/6.2 mile run. Of note, sprint-distance races were not included. Events, where the standard distance was altered due to technical or environmental issues (i.e., reduced course; excluded swim split), were excluded from the analysis. In addition, drafting became officially legal in the year of 2000 [20], so we also grouped the events according to drafting rules for analysis (1989 to 1999 vs. 2000 to 2019).

Men and women were included and analyzed independently and compared when pertinent. The total sampling included 52,027 entries, 33,099 of which were men and 18,928 women, from 1191 WTS race events and four Olympic race events. All race data (i.e., overall race time, swim, cycle and run split times) were converted to minutes for data analysis.

### 2.3. Data Analysis

Continuous data had normality and homogeneity assessed with Kolmogorov–Smirnov’s and Levene’s tests, respectively. Since all continuous variables presented parametric distributions, data were expressed as mean and standard deviation (μ ± SD). Linear regressions were applied for performance trends. Performance trends with just the top three athletes of each race in a given year were also examined. A Student’s t-test for independent samples was applied to compare men and women. With overall race time as the dependent variable, multiple regression analyses were performed to assess which split (independent variable) may have the greater influence. In addition, sensitivity, specificity, positive and negative likelihood ratios, odds ratios, and confidence intervals (95%) for the podium position, considering the cut-off value (Receiver Operating Characteristic [ROC] curve) for the top three rank positions, were calculated. The significance level was 5% (*p* < 0.05). All analyses were performed using Statistical Software for the Social Sciences (SPSS v21.0, Chicago, IL, USA) and GraphPad Prism (Prism v6.0, San Diego, CA, USA).

## 3. Results

Men were significantly faster than women in all splits (swim: 19.7 ± 2.5 min vs. 21.1 ± 2.3 min; cycle: 61.5 min ± 5.7 vs. 68.3 ± 6.1 min; run: 35.7 ± 4.0 min vs. 40.0 ± 4.1 min) and consequently in overall race time (117.7 ± 9.4 min vs. 130.4 ± 9.7 min). Linear regressions showed a negative slope for both men and women in overall times across the years from 1989 to 2018 (Figure 1).

Split times of the top three athletes in each race were compared between nondraft and draft years, for both men and women. Men and women were faster swimmers during the drafting period (18.8 ± 1.4 min and 20.6 ± 2.0 min, respectively) in comparison to nondraft (19.1 ± 2.0 min and 21.0 ± 2.0 min). In cycling, both sexes became slower in the drafting period (men: 60.3 ± 4.6 min vs. 58.7 ± 5.2 min; women: 67.5 ± 5.4 min vs. 65.3 ± 5.4 min). No significant differences were identified for the running split (Figure 2).

Multivariate regression analysis showed that cycling was the discipline with the lowest influence on overall race time for both men and women, and with the sample divided in top three, ≥4th place, or altogether (Table 1). Running was the discipline with the highest influences on overall race times for men regardless of rank position. For women, swimming was the discipline with the highest influences on overall race time only when ranked ≥ 4th place. The multivariate regression carried out on different decades showed that cycling had the lowest influence on overall race time for both men and women in all three decades, with the exception of 2009–2018, where swimming showed the lowest influence followed by cycling (Table 1).

The cut-off values to achieve a top three ranking obtained by the overall race times were 114.8 and 126.8 min for men and women, respectively. Furthermore, the cut-off values stratified by sex in each discipline to achieve overall a top three ranking were also significant. Men had cut-off values as follows—swimming = 19.5 min (average pace; AP: ~1:18 min/100 m; ~1:11 min/100 yrd); cycling 61.6 min (average speed; AS: ~39.5 km/h; ~24.6 mph); running = 34.1 min (AP: ~3:25 min/km; ~5:29 min/mile) (Figure 3). Women had cut-off values as follows with swimming = 20.7 min (AP: ~1:23 min/100 m; ~1:16 min/100 yrd); cycling 71.6 min (AS: ~33.5 km/h; ~20.8 mph); running = 38.1 min (AP: ~3:49 min/km; ~6:08 min/mile) (Figure 4).

Men with a performance below the overall cut-off points presented a significantly higher possibility of achieving a podium position when compared to the reference (above cut-off point) group (odds ratio: 3.453 (CI: 95%; 3.163–3.769)). Men’s cut-off point proposed for overall race time demonstrated high sensitivity but not a high specificity to predict a top three performance. Likewise, women performing below the overall cut-off points showed a higher chance of achieving a top three position in a standard Olympic distance triathlon (odds ratio: 2.211 (CI: 95%; 2.035–2.402)). The women’s cut-off point proposed for overall race time demonstrated high specificity but not a high sensitivity to predict a top three performance.

Furthermore, the odds ratios of cut-off points for men in each discipline were 2.074 (95% CI: 1.899–2.266), 1.575 (95% CI: 1.452–1.709), and 6.629 (95% CI: 6.018–7.301) for swimming, cycling and running, respectively. For women, it was 1.382 (95% CI: 1.272–1.50), 1.449 (95% CI: 1.305–1.609), and 3.581 (95% CI: 3.289–3.898) for swimming, cycling, and running, respectively.

## 4. Discussion

The main results of this study were: (i) unprecedented cut-off points established for a successful race in an Olympic distance triathlon for elite men and women; (ii) the running split seemed to be the most influential in overall race time regardless of rank position or sex; (iii) cycling was the discipline with the lowest influence on overall race time; (iv) the influence of cycling on overall results increased with drafting rules; (v) in nondrafting years, swimming was, on average, slower and cycling faster when compared to drafting years.

Running was the most important split discipline for overall race performance. Regarding the split disciplines, running is the final discipline after cycling and, therefore, strongly influenced by the preceding discipline, cycling [21,22]. It has been shown that aspects such as cycling cade [23,24] and position in the cycling split [25] had an influence on running performance. Furthermore, leg muscle activity during running is influenced by cycling [26,27].

Men showed to be faster swimmers, cyclists, runners, and, consequently, faster triathletes than women. This outcome was previously reported for open-water swimmers [28], recreational and elite marathoners [29,30], and triathletes competing in an Ironman triathlon [31]. Due to sex-related gene expression and their interactions with sex hormones [32,33], the skeletal muscles of men are larger and may have a metabolically/functionally greater area of muscle fibers than women [34,35], leading to the aforementioned improved performance. Although women seem to have less fatigability than men [36] and research has shown that they may even be closing the performance gap in ultra-endurance races [31,37], it seems men are still faster in Olympic distance triathlons for the time being.

Linear regressions showed that cycling was the split discipline with the lowest influence on overall performance in an Olympic distance triathlon for both men and women, and that this pattern was similar in the three decades of the WTS circuit. This result corroborates previous studies investigating Olympic distance triathlons [2,4], but was contrary to similar analyses performed for Ironman distance triathlons [8,10]. Race regulations regarding drafting in cycling lead to this outcome, even though cycling comprises a significant percentage of overall race time in every triathlon race distance. Utilizing the draft or slipstream created by the cycling peloton allows the athlete to go faster while saving energy for the running split [3,4,7]. Since drafting is allowed in Olympic distance triathlons, athletes usually cycle the uniform peloton formation, leaving most of the competitiveness to the running discipline, which was the most influential in overall race performance [2].

The change in influence of cycling on overall performance throughout the years may have changed the strategy pattern for triathlons as the decades went by. Overall, race time became faster throughout the years, as well as swimming and running times, but not cycling. Piacentini et al. [2] reported that finishing the swimming split and running close to the first athlete was a major determinant for success, indicating that male athletes should be close to the leaders when leaving transition two (cycle-to-run) to succeed. Indeed, the running split is the most influential on overall performance in men in the present analysis.

Furthermore, the swimming split seemed to have an even greater influence on performance in women. Research showed that women were less able to bridge cycle packs than men [3], making the swimming split even more competitive since it is harder to reach the leaders if you are about to cycle in a chasing pack [2,38]. Accordingly, the present results showed that the swimming split was the discipline with the greatest influence on overall race performance in Olympic distance triathlons for women ranked ≥ 4th place or altogether, but not for the top three. For the top three positions, the running split remains the most important, and more than ever, it seems that, for the women’s race, one must be a very fast swimmer to compete and an even faster runner to win.

Although the present study shows advances in strategies to achieve success in the Olympic triathlon, aspects such as climatic conditions and race courses [7] were not considered in the present analysis, which is one of the limitations of this study. On the other hand, it is worth mentioning that the information publicly available is the environment and water temperature for races after 2010. Thus, we suggest that race organizers report more information after each race (such as altimetry gain, wind, etc.) so that future research could consider those variables in the analysis to test how much it can affect the athlete’s performance. A second limitation of the study is that technological advancement in equipment since 1989 may also have influenced race times. In this context, wetsuits, bicycles, and running shoes may have improved race times. However, this analysis is beyond the scope of the current study, but would make for an interesting topic in future studies. Additionally, drafting and nondrafting have unequal sample sizes, which could potentially bias the comparison results. Indeed, it is surprising that cycling became slower with legal drafting. There are much more legal-drafting events than nondrafting events, which caused the unequal sample size. Although nondrafting events were faster (in average), the fastest splits in cycling were in the drafting years.

The main contribution of the present study is its novelty. As far as we know, there are no reports in the literature of studies that investigated cut-off points for a successful race in the Olympic distance triathlon for elite men and women. In addition, no analysis of the influence of each discipline on the overall race time was made with all available data, which is more than 50,000 athletes. Thus, this knowledge is unique and can have extensive applications within the Olympic distance triathlon.

The present study has several practical applications. The cut-off points set for both men and women for all three disciplines may be a more significant practical application parameter for athletes and coaches. Achieving a race pace better than the cut-off point (or race time below) means a greater chance (3.5 to men and 2.2 for women) of achieving a top three rank position in an Olympic distance triathlon. It is noteworthy that environment, course, and rivalry could be more determinant of an athlete’s race strategy than a pace goal [3]. Applying a basic concept previously stated by Hanley and Hettinga [39] for middle-distance Olympic runners that “Champions are racers, not pacers”. Nevertheless, for emerging triathletes (juniors, under 23 years/amateurs willing to professionalize), the cut-off points may be of use to establish seasonal training goals or identify potential talents.

Although the strength of the study is the long time frame and the high number of considered athletes, variables such as training [11] and physiological characteristics [40] are missing. Furthermore, aspects with an influence on race performance such as the transition between the split disciplines [41] and nutrition during the race [42] have not been considered.

## 5. Conclusions

In conclusion, the established cut-off points were set to increase the chances of achieving a successful rank position in an Olympic triathlon (WTS and Olympics). It is noteworthy that these cut-offs just represent a guide for training routine of Olympic distance triathletes. Obviously, each triathlon race presents a specificity that needs to be carefully analyzed. Furthermore, cycling is the discipline with the least influence on overall performance for both men and women over the Olympic distance, while swimming has a greater influence on performance for women and running for men. This influence pattern has not changed in the last three decades of the WTS circuit. Further research should consider including environmental variables in their analyses to assess how the weather (e.g., wind, rain, temperature, humidity) or course peculiarities (e.g., altimetry gain, sharp curves, etc.) could affect performance and how elite triathletes shift their strategies accordingly.

## Figures and Tables

**Figure 1 ijerph-17-09491-f001:**
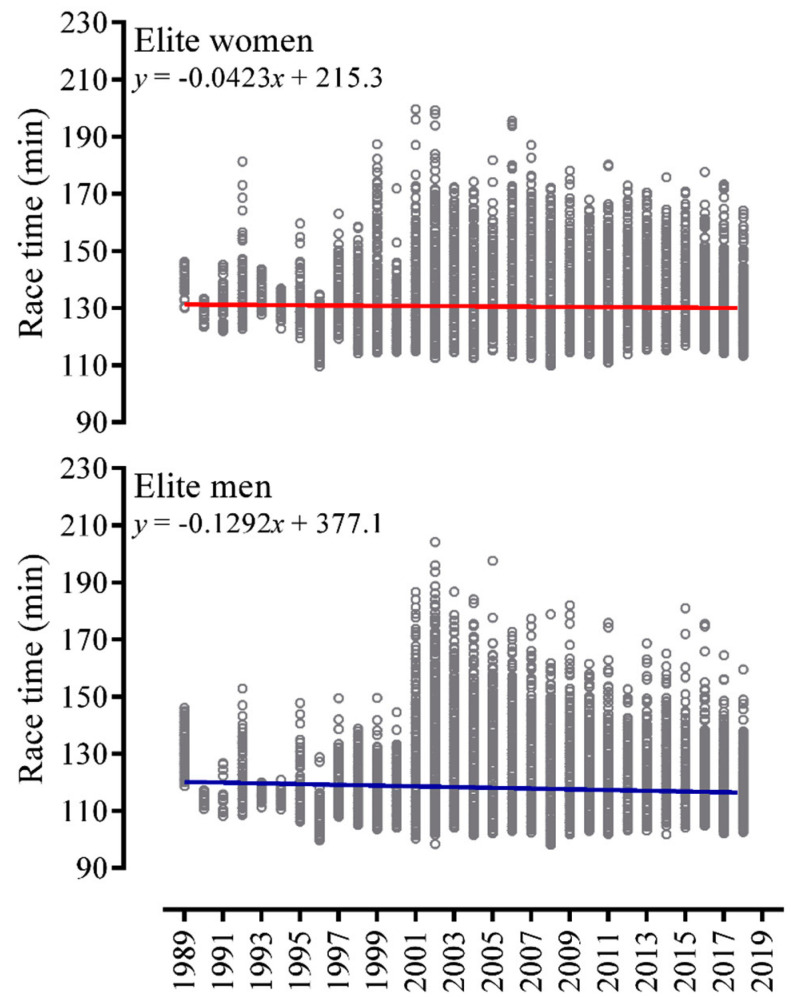
Performance trend of overall race time in Olympic distance triathlon from 1989 to 2019 in men and women.

**Figure 2 ijerph-17-09491-f002:**
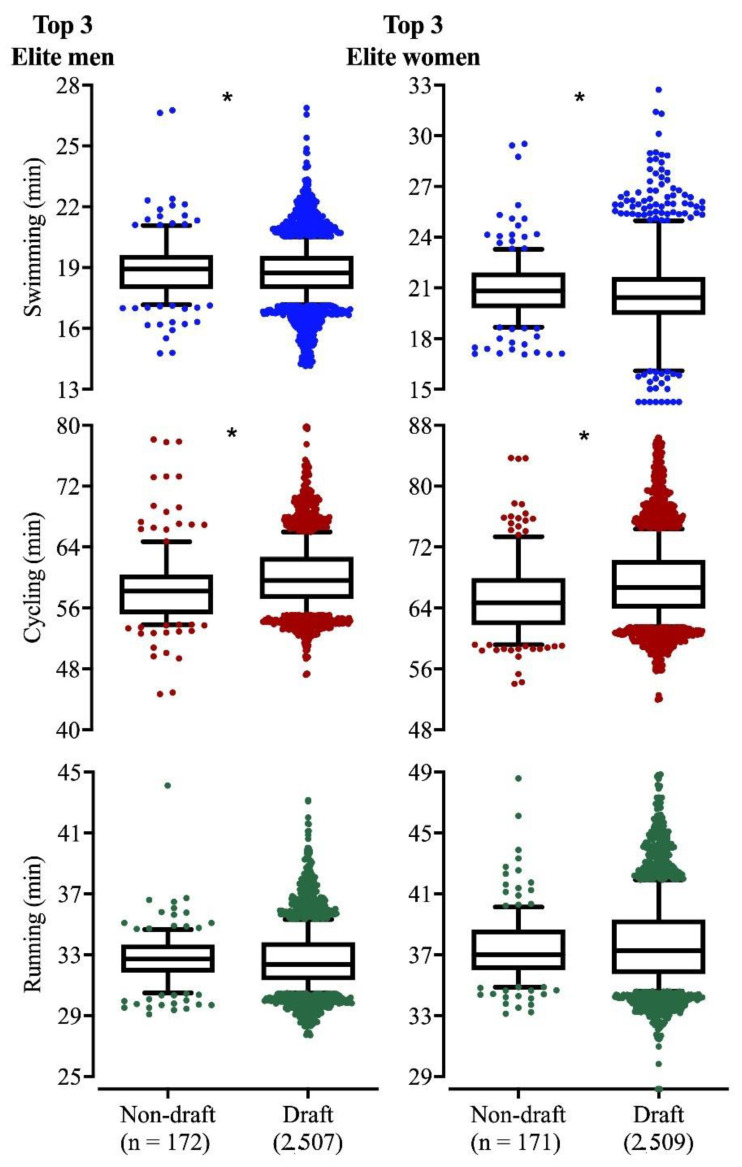
Average race times of each discipline (swim; cycle; run) of the top three finishers between draft (2000–2019) and nondraft periods (1989–1999). *: *p* < 0.05.

**Figure 3 ijerph-17-09491-f003:**
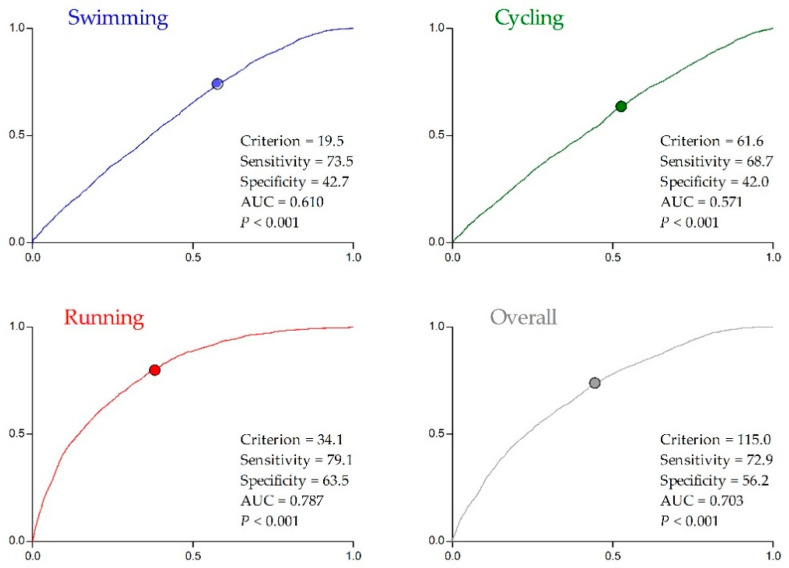
Receiver operating characteristic curve (sensitivity: *y*-axis; specificity: *x*-axis) according to the top three male rank position of each trait of triathlon (Draft period: 2000–2019): AUC = area under the curve.

**Figure 4 ijerph-17-09491-f004:**
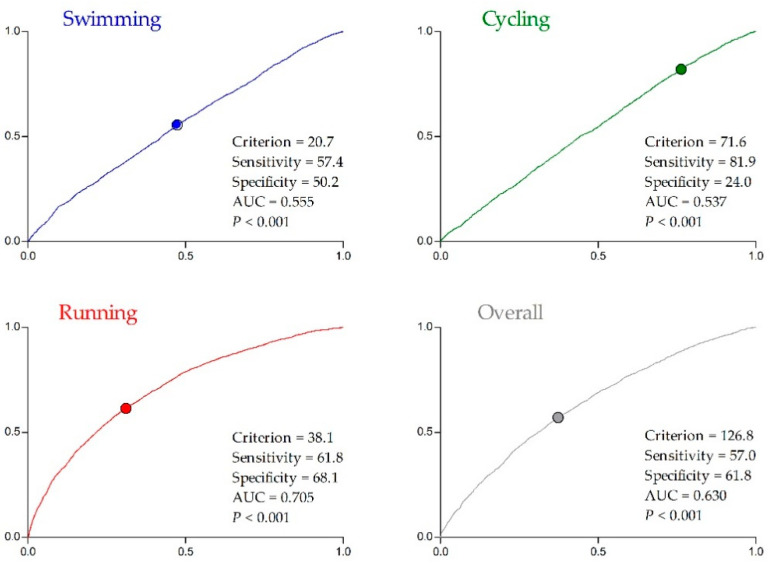
Receiver operating characteristic curve (sensitivity: *y*-axis; specificity: *x*-axis) according to the top three female rank positions of each trait of triathlon (Draft period: 2000–2019): AUC = area under the curve.

**Table 1 ijerph-17-09491-t001:** Multivariate regression results to determine discipline influence in Olympic triathlon altogether and clustered by race ranking and between decades with and without draft rules.

		R	R^2^	*p*-Value	Coefficients		
					Swim	Cycle	Run
Race ranking							
Elite men	All (*n* = 33,098)	0.981	0.962	<0.001	1.030	0.884	1.035
	Top 3 (*n* = 2678)	0.952	0.906	<0.001	0.971	0.855	0.975
	≥4th (*n* = 30,418)	0.981	0.963	<0.001	1.030	0.886	1.034
Elite women	All (*n* = 18,928)	0.991	0.982	<0.001	1.012	0.966	1.005
	Top 3 (*n* = 2680)	0.987	0.975	<0.001	0.984	0.971	1.019
	≥4th (*n* = 16,247)	0.991	0.982	<0.001	1.015	0.966	1.001
Nondraft (1989–1999) vs. draft (2000–2019)							
Elite men	Nondraft (*n* = 2269)	0.955	0.912	<0.001	1.044	0.780	1.080
	Draft (*n* = 30,830)	0.982	0.965	<0.001	1.032	0.891	1.029
Elite women	Nondraft (*n* = 1841)	0.957	0.916	<0.001	1.083	0.801	1.093
	Draft (*n* = 17,087)	0.995	0.990	<0.001	1.001	0.988	0.994

Overall race time (min) was used as a dependent variable for all analyses; Top 3: the top three athletes in a single event; ≥4th: all athletes in a single event that finished fourth and below.

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
