# Peer review of "Cut-Off Values in the Prediction of Success in Olympic Distance Triathlon"

_ijerph, 2020, doi:10.3390/ijerph17249491_

Round 1
Reviewer 1 Report
The authors have produced a lot of interesting data which deserves publication and I have no major criticisms.
The main result of this manuscript ("Cut-off Values in the Prediction of Success in Olympic Distance Triathlon") was unprecedented cut-off points established for a successful race in an Olympic distance triathlon for elite men and women in the last decade. The total sampling included 52,027 entries, 33,099 of which were men and 18,928 women, from 1,191 World Triathlon Series race events and four Olympic race events. Although the present study shows advances in strategies to achieve success in the Olympic triathlon, aspects such as environmental factors (e.g. climatic conditions) and technological advancement (e.g. race courses) were not considered in the present analysis. However the authors explain these limitations in the article. The strong points of the article are: • the topic of the article clearly defined; • the text well clearly written in an orderly way; • the evidence accurate and it sufficient to support the points being argued; • the evidence presented appropriate for the intended audience (sports scientists, professional triathletes and their coaches); • the arguments of the article leads to possible applications of the theory and would be an important factors for athletes planning, training schedules and competition strategies; • The main contribution of the present study is its novelty. There are limited reports in the literature of studies that investigated cut-off points for a successful race in the Olympic distance triathlon for elite men and women. In addition, authors provide analysis of the influence of each discipline on the overall race time was made with all available data (more than 50 thousand athletes). Thus the authors have produced a lot of interesting and scientific data which definitely deserves publication.
Author Response
Reviewer #1 comments
The authors have produced a lot of interesting data which deserves publication and I have no major criticisms. The main result of this manuscript ("Cut-off Values in the Prediction of Success in Olympic Distance Triathlon") was unprecedented cut-off points established for a successful race in an Olympic distance triathlon for elite men and women in the last decade. The total sampling included 52,027 entries, 33,099 of which were men and 18,928 women, from 1,191 World Triathlon Series race events and four Olympic race events. Although the present study shows advances in strategies to achieve success in the Olympic triathlon, aspects such as environmental factors (e.g. climatic conditions) and technological advancement (e.g. race courses) were not considered in the present analysis. However, the authors explain these limitations in the article. The strong points of the article are: • the topic of the article clearly defined; • the text well clearly written in an orderly way; • the evidence accurate and it sufficient to support the points being argued; • the evidence presented appropriate for the intended audience (sports scientists, professional triathletes and their coaches); • the arguments of the article leads to possible applications of the theory and would be an important factors for athletes planning, training schedules and competition strategies; • The main contribution of the present study is its novelty. There are limited reports in the literature of studies that investigated cut-off points for a successful race in the Olympic distance triathlon for elite men and women. In addition, authors provide analysis of the influence of each discipline on the overall race time was made with all available data (more than 50 thousand athletes). Thus, the authors have produced a lot of interesting and scientific data which definitely deserves publication.
Response: Dear Reviewer, we appreciate the support of our study.
Reviewer 2 Report
Review for IJERPH
Title: Cut-off Values in the Prediction of Success in Olympic Distance Triathlon
General comments:
The aims of the paper are very practical. The purposes of the present study were (i) to determine cut-off values to reach the top three position in Olympic distance triathlon; and (ii) to identify which discipline present the highest influence on overall race performance and if it has changed over the decades.
After reading the whole thing, I wonder if it would not be better not to take into account only the first 3 competitors, but for example the top ten - but this is just my private feeling which please do not take into account.
Congratulations to the authors, the article is very interesting. Requires minor modifications, including extension of the introduction. I suggest publishing the article with minor corrections
Specific comments:
Introduction
In the introduction, the authors focus on only one discipline of triathlon, i.e. cycling. It seems to add a paragraph about swimming and running
At the end of the introduction, a good justification for undertaking such an analysis.
Materials and Methods
A very big advantage of these studies is the very large number of participants
Is this data publicly available? Can the authors use this data for themselves? Do they have permission to do so?
Results
Tables and figures legible
Discussion
The discussion is well written. In the final paragraphs, the practical application and limitations of my research were presented, which are highly mature and highly competent authors
Conclusions
Supported by results
References
Good selection of literature. I would consider deleting some of the oldest
Author Response
Reviewer #2 comments
General comments: The aims of the paper are very practical. The purposes of the present study were (i) to determine cut-off values to reach the top three position in Olympic distance triathlon; and (ii) to identify which discipline present the highest influence on overall race performance and if it has changed over the decades.
After reading the whole thing, I wonder if it would not be better not to take into account only the first 3 competitors, but for example the top ten - but this is just my private feeling which please do not take into account.
Congratulations to the authors, the article is very interesting. Requires minor modifications, including extension of the introduction. I suggest publishing the article with minor corrections
Response: Thank you for the support of our paper.
Specific comments:
Introduction
In the introduction, the authors focus on only one discipline of triathlon, i.e. cycling. It seems to add a paragraph about swimming and running
Response: From your suggestion, we emphasized that there is individual importance of each discipline for the overall-race success in the triathlon, thus that afterwards cycling is presented as a pivotal discipline in some events of triathlon, such as Ironman.
At the end of the introduction, a good justification for undertaking such an analysis.
Response: We added a justification for analysis at the end of the Introduction.
Materials and Methods
A very big advantage of these studies is the very large number of participants
Is this data publicly available? Can the authors use this data for themselves? Do they have permission to do so?
Response: The paragraph that addresses these issues (section 2.2) has been rewritten to provide all the information requested by the Reviewer#2, as following “All data were sourced from the official, publicly available results of the World Triathlon Series (WTS) events from 1989 to 2019 (www.triathlon.org/results) and triathlon races held during the Olympic Games (2004 to 2016) (www.triathlon.org/olympics/history). In order to obtain the results efficiently, in a standardized fashion and with minimized human error, a custom python script was written and used to download the desired events. The standard distance for those events was 1.5km/0.93 mile swim, 40 km/24.9 mile cycle, and 10 km/6.2 mile run. Events, where the standard distance was altered due to technical or environmental issues (i.e. reduced course; excluded swim split), were excluded from the analysis.”.
Results
Tables and figures legible
Response: No further changes are required.
Discussion
The discussion is well written. In the final paragraphs, the practical application and limitations of my research were presented, which are highly mature and highly competent authors
Response: No further changes are required.
Conclusions
Supported by results
Response: No further changes are required.
References
Good selection of literature. I would consider deleting some of the oldest
Response: We appreciate the suggestion. Sincerely, we believe that all the references cited were necessary to concatenate the ideas presented during the text and also to discuss/explain the findings of our study. Thank you.
Reviewer 3 Report
This research articles examines the cut off times for placing in the top 3 at World event in Olympic distance triathlon, and how performance in each event would contribute to a podium finish. The results of the past 30 years of data was examined.
This is as the authors mention a unique study which can be used to educate triathlon coaches on training focus to successfully reach the podium on the triathlon world stage. The one weakness of the study is examining cut off times of these races and trying to determine what may place in future races. As the authors noted in their limitations that comparing races over this time frame and locations of races does not account for the race course/terrain, weather, etc. I am not sure we can conclude that the information they found here would be valid to apply to the general population since so many factors go into racing. I would consider revising this research question and perhaps instead presenting it as descriptive statistics, as opposed to determining cut off values that would be used by coaches for training purposes.
Methods Section: I feel that your methods sections is lacking information. Granted your data collection was based on gathering data from an established data base. There is no procedures section listed, only a sample and data analysis section. Although I understand why this may be the case.
Discussion Section: You mention that there are gender differences in the results which would be an expected outcome. It has been well established that males out perform females during elite races. You confirm this with previous studies. I feel that you should mention that this would be typically seen, and connect back to physiologically why this is the case.
I also found it interesting that cycling times decreased. As you mentioned improvements in equipment should see the opposite occur. I would look into some articles to see if any other studies have found this, and why this may be the case.
Author Response
Reviewer #3 comments
This research articles examines the cut off times for placing in the top 3 at World event in Olympic distance triathlon, and how performance in each event would contribute to a podium finish. The results of the past 30 years of data was examined.
This is as the authors mention a unique study which can be used to educate triathlon coaches on training focus to successfully reach the podium on the triathlon world stage. The one weakness of the study is examining cut off times of these races and trying to determine what may place in future races. As the authors noted in their limitations that comparing races over this time frame and locations of races does not account for the race course/terrain, weather, etc. I am not sure we can conclude that the information they found here would be valid to apply to the general population since so many factors go into racing. I would consider revising this research question and perhaps instead presenting it as descriptive statistics, as opposed to determining cut off values that would be used by coaches for training purposes.
Response: Although it has already been highlighted as a limitation, we added a sentence in the conclusion to consider the comment of Reviewer # 3. Additionally, by these cutoffs, there is no pretension to predict what will happen in future of Olympic Distance triathlon races, but to generate parameters based on consistent results (over 30 years) for the training schedule of athletes. Of note, the aforementioned parameters are very important for a coach’s decision; obviously, respecting the specificities of each race, as the Reviewer # 3 mentioned (e.g., course/terrain and weather).
Methods Section: I feel that your methods sections is lacking information. Granted your data collection was based on gathering data from an established data base. There is no procedures section listed, only a sample and data analysis section. Although I understand why this may be the case.
Response: Dear reviewer, as you mentioned, the procedures of the present study are data selection, data processing, and data analysis. We believe that all possible information in that regard is fully described in the methods. We further added more information considering yours and other reviewer’s specific suggestions. If there is any other information that may happen to be missing, we will be happy to add it. Thank you for your comment.
Discussion Section: You mention that there are gender differences in the results which would be an expected outcome. It has been well established that males outperform females during elite races. You confirm this with previous studies. I feel that you should mention that this would be typically seen, and connect back to physiologically why this is the case.
Response: In attempt to provide more details, the paragraph that addresses these issues (section 2.2) has been rewritten, as follows “All data were sourced from the official, publicly available results of the World Triathlon Series (WTS) events from 1989 to 2019 (www.triathlon.org/results) and triathlon races held during the Olympic Games (2004 to 2016) (www.triathlon.org/olympics/history). In order to obtain the results efficiently, in a standardized fashion and with minimized human error, a custom python script was written and used to download the desired events. The standard distance for those events was 1.5km/0.93 mile swim, 40 km/24.9 mile cycle, and 10 km/6.2 mile run. Events, where the standard distance was altered due to technical or environmental issues (i.e. reduced course; excluded swim split), were excluded from the analysis.”.
I also found it interesting that cycling times decreased. As you mentioned improvements in equipment should see the opposite occur. I would look into some articles to see if any other studies have found this, and why this may be the case.
Response: To the best of our knowledge, no studies that provided the aforementioned information for Olympic distance triathlon were found. Thus, it becomes difficult to mention some work who has similar results. In this sense, we believe that it will be a starting point for furthers investigations.
Reviewer 4 Report
The authors have done a very good job. It is really surprising that there were no cut-off values to achieve performance in Olympic distance triathlon. They have downloaded all the registration data and the work consists of 3 decades of tracking, with 57,027 entries (> 33,000 male and almost 20,000 female) from 1,191 WTS races and four Olympics. The practical application of the work is, in addition to being obvious, important: to provide reference values to the coaches on what times their athletes must reach (in each discipline) to have the best chance of success. In addition, the relative importance of each discipline contributes to the possibility of success, which also allows deciding how much to dedicate to training each one of them.
In my opinion it is well written and methodologically correct. I can't find correctable aspects except for two little things:
Sample (lines 93-95). Would a comparison between the WTS and the Olympics events be possible? Despite the more than obvious difference in the sample, perhaps it would allow us to detect some difference in the study parameters that would differentiate these two types of competition.
Lines 126-127. Figure 2 legend: please include "of the top three winners"
Author Response
Reviewer #4 comments
The authors have done a very good job. It is really surprising that there were no cut-off values to achieve performance in Olympic distance triathlon. They have downloaded all the registration data and the work consists of 3 decades of tracking, with 57,027 entries (> 33,000 male and almost 20,000 female) from 1,191 WTS races and four Olympics. The practical application of the work is, in addition to being obvious, important: to provide reference values to the coaches on what times their athletes must reach (in each discipline) to have the best chance of success. In addition, the relative importance of each discipline contributes to the possibility of success, which also allows deciding how much to dedicate to training each one of them.
In my opinion it is well written and methodologically correct. I can't find correctable aspects except for two little things:
Sample (lines 93-95). Would a comparison between the WTS and the Olympics events be possible? Despite the more than obvious difference in the sample, perhaps it would allow us to detect some difference in the study parameters that would differentiate these two types of competition.
Response: Dear Reviewer # 4, we appreciate the support of our study, and the opportunity to clarify some aspects of the manuscript. Regarding to the suggestion of a comparison between WTS and Olympics events, it is an extremely relevant question. However, there are data from only four Olympic events, whereas WTS events have >10 events/year. Additionally, this comparison does not fall within the scope of our current aims. Nevertheless, we appreciate the suggestion and we will consider it for a further analysis/manuscript of our research group. Thank you for this valuable insight!
Lines 126-127. Figure 2 legend: please include "of the top three winners"
Response: The following information has been added “of the top three finishers”.
Reviewer 5 Report
I appreciate the effort of author preparing this manuscript. Also under my opinion this paper might be very interesting in triathlon world.
However, the paper could improve. I send you my suggestions:
ABSTRACT:
It may include some introductory lines, not just start with aims.
What is “official Olympic distance” . It refers international, ITU events, World cup…?
INTRODUCTION
- Include a reference for important of bike segment in Ironman Triathlon (line 52)
- Include were drafting was allowed in triathlon. In 80’s and early 90’s it was allowed and perhaps data interpretation may change.
- I guess that the study refers to ITU events (world champs, world cups, etc.). It should be clarified, because it may induce to think about “all” Olympic distance triathlons… (nationals, regionals…)
METHODS
- In latest years, some sprint distance events has been included in WTS. Have these data been included in the paper?
- If it is not, I suggest to included them, extend the paper and analyze if there are differences between Olympic and sprint distance in WTS.
- First year of race (1989-middle 90’s) triathlon races were without drafting. Please, exclude these data from the analysis and compare them with drafting events. Also, first decade could be changed by “no-drafting” period. I guess that splits time will be much different, specially swim and run.
- A summary of number of races analyzed per year (or decade) and kind (WTS – World Cup, World Champ…) might be interesting for a better understanding.
RESULTS
- Results presentation should be adjusted to new proposal about methodology:
- Include sprint triathlons?
- Include “other” international events (world cups, Etu events…)
- No-drafting “decade” vs Drafting decades.
DISCUSSION
- This section is proper but it may change with new data and changes in methodology. For example, I guess that there is difference in cycling segment between WTS and “lower level” races (ETU cups).
- As cut offs points is one of the main practical application of the research, the should be more detailed in the document, at least last decade. So coaches may have the reference. Also set differences for Olympics Games, WTS, World cups…
- Line 235. It is written that strength of the study is long time frame, 11 years, but study includes races from 1989 to 2018, so …29 years?
CONCLUSIONS
- At the beginning of this section, it must be clarified that conclusion are for a determined level of Olympic distance triathlons (WTS – ITU, etc.) not for all Olympic distance triathlons.
Author Response
Reviewer #5 comments
I appreciate the effort of author preparing this manuscript. Also, under my opinion this paper might be very interesting in triathlon world. However, the paper could improve. I send you my suggestions:
Response: Based on your insights, the manuscript underwent a review that made it more suitable for publication. Please find below our responses to each of your specific comment. Thank you.
ABSTRACT:
It may include some introductory lines, not just start with aims.
Response: Done.
What is “official Olympic distance” . It refers international, ITU events, World cup…?
Response: Olympic distance triathlon has presented in the introduction section as: 1.5 km/0.93 mile swim; 40 km/24.9 mile cycle; and 10 km/6.2 mile run. Unfortunately, due to the limited number of words, there is not enough space to clarify these points in the Abstract.
INTRODUCTION
Include a reference for important of bike segment in Ironman Triathlon (line 52)
Response: Done.
Include were drafting was allowed in triathlon. In 80’s and early 90’s it was allowed and perhaps data interpretation may change.
Response: Dear reviewer, drafting is an official rule the varies from country to country (federation to federation) for age-groupers and indeed, has been changing along the years. However, drafting was allowed for professional/elite triathletes in official ITU, WTS events, and Olympics. Therefore, drafting rule is not cannot be considered and tested as an independent variable considering the current dataset (elite athletes racing Olympic-distance official events).
METHODS
In latest years, some sprint distance events has been included in WTS. Have these data been included in the paper? If it is not, I suggest to included them, extend the paper and analyze if there are differences between Olympic and sprint distance in WTS.
Response: The following paragraph has been rewritten to clarify our sample “All data were sourced from the official, publicly available results of the World Triathlon Series (WTS) events from 1989 to 2019 (www.triathlon.org/results) and triathlon races held during the Olympic Games (2004 to 2016) (www.triathlon.org/olympics/history). In order to obtain the results efficiently, in a standardized fashion and with minimized human error, a custom python script was written and used to download the desired events. The standard distance for those events was 1.5km/0.93 mile swim, 40 km/24.9 mile cycle, and 10 km/6.2 mile run. Events, where the standard distance was altered due to technical or environmental issues (i.e. reduced course; excluded swim split), were excluded from the analysis.”.
First year of race (1989-middle 90’s) triathlon races were without drafting. Please, exclude these data from the analysis and compare them with drafting events. Also, first decade could be changed by “no-drafting” period. I guess that splits time will be much different, specially swim and run. A summary of number of races analyzed per year (or decade) and kind (WTS – World Cup, World Champ…) might be interesting for a better understanding.
Response: Dear reviewer, our current goal with this manuscript was to establish the cut-off points for Olympic-distance triathlon. Such analysis required a good and representative sample size. We believe that Sprint-distance triathlon needs further development and more events before such analysis can make an impact. Nevertheless, we appreciate the suggestion, and we will consider this analysis in our future research plans. Thank you for this insight!
RESULTS
Results presentation should be adjusted to new proposal about methodology:
Include sprint triathlons?
Include “other” international events (world cups, Etu events…)
No-drafting “decade” vs Drafting decades.
Response: Please see response to previous comments.
DISCUSSION
This section is proper but it may change with new data and changes in methodology. For example, I guess that there is difference in cycling segment between WTS and “lower level” races (ETU cups).
Response: Dear reviewer, we appreciate your effort to review our manuscript. Some of the analysis you suggested will be considered in our future investigations.
As cut offs points is one of the main practical application of the research, they should be more detailed in the document, at least last decade. So, coaches may have the reference. Also set differences for Olympics Games, WTS, World cups…
Response: The present study has provided cutoffs for all events (WTS and Olympics). Thus, as suggested by reviewer # 5, we´ll consider specific cutoff points for each kind of event in further studies. Thank you for this insight.
Line 235. It is written that strength of the study is long time frame, 11 years, but study includes races from 1989 to 2018, so …29 years?
Response: There was a typo in this sentence. By this time, it has adequately rewritten.
CONCLUSIONS
At the beginning of this section, it must be clarified that conclusion are for a determined level of Olympic distance triathlons (WTS – ITU, etc.) not for all Olympic distance triathlons.
Response: Done.
Round 2
Reviewer 5 Report
Dear authors, the paper has improved, but under my knowledge, it still needs some important changes.
ABSTRACT:
Please indicate in the abstract that “official Olympic distance” refers to WTS and Olympics. Word limit is not a problem. Rewrite some sentences.
INTRODUCTION
It is important to know the evolution of triathlon and “races analyzed” during the period of the study.
From 1989 (first Triathlon World Championship) to 2008, the world championship was a single race. In 2009 WTS were created and World Championship was based on several triathlon races and a Great Final.
Furthermore, in triathlon world championships until middle 90’s drafting was not allowed, so cut off times maybe different from drafting races.
This information must appear in the introduction.
METHODS
“Response: The following paragraph has been rewritten to clarify our sample “All data were sourced from the official, publicly available results of the World Triathlon Series (WTS) events from 1989 to 2019 (www.triathlon.org/results) and triathlon races held during the Olympic Games (2004 to 2016) (www.triathlon.org/olympics/history). In order to obtain the results efficiently, in a standardized fashion and with minimized human error, a custom python script was written and used to download the desired events. The standard distance for those events was 1.5km/0.93 mile swim, 40 km/24.9 mile cycle, and 10 km/6.2 mile run. Events, where the standard distance was altered due to technical or environmental issues (i.e. reduced course; excluded swim split), were excluded from the analysis.”.
That is right, but still it is not clear if official sprint races included in WTS were analyzed or not. They did not were altered due to technical or environmental issues.
“Response: Dear reviewer, our current goal with this manuscript was to establish the cut-off points for Olympic-distance triathlon. Such analysis required a good and representative sample size. We believe that Sprint-distance triathlon needs further development and more events before such analysis can make an impact. Nevertheless, we appreciate the suggestion, and we will consider this analysis in our future research plans. Thank you for this insight!”
I agrre, but If this races were excluded, please include a table(or lines) indicating year by year the number of races analyced per year and participants, male and female. It will give more information to the reader and will help to understand better the paper and data.
Please consider to organize data is I suggest as follows.
RESULTS
“Please, reconsider organize your data (table 1), not by decades. At least 1989-.midle 90’s (no drafting period), middle 90’s – 2008 (World Championship for a single race) 2009-2019 (WTS). This should be indicated in methods section.
Triathlon races was “different” in these 3 period of time.
DISCUSSION
Please, consider include discuss results and cut off time divided by those 3 periods of time.
This paper should be also included in discussion section.
Ofoghi B, Zeleznikow J, Macmahon C, Rehula J, Dwyer DB. Performance analysis and prediction in triathlon. J Sports Sci. 2016;34(7):607-12. doi: 10.1080/02640414.2015.1065341. Epub 2015 Jul 16. PMID: 26177783.
Author Response
Reviewer #5 comments
Dear authors, the paper has improved, but under my knowledge, it still needs some important changes.
Response: Dear Reviewer, we appreciate the support of our study. Efforts were demanded to improve the paper. Thank you!
ABSTRACT:
Please indicate in the abstract that “official Olympic distance” refers to WTS and Olympics. Word limit is not a problem. Rewrite some sentences.
Response: Done.
INTRODUCTION
It is important to know the evolution of triathlon and “races analyzed” during the period of the study.
From 1989 (first Triathlon World Championship) to 2008, the world championship was a single race. In 2009 WTS were created and World Championship was based on several triathlon races and a Great Final.
Furthermore, in triathlon world championships until middle 90’s drafting was not allowed, so cut off times maybe different from drafting races.
This information must appear in the introduction.
Response: Done.
METHODS
“Response: The following paragraph has been rewritten to clarify our sample “All data were sourced from the official, publicly available results of the World Triathlon Series (WTS) events from 1989 to 2019 (www.triathlon.org/results) and triathlon races held during the Olympic Games (2004 to 2016) (www.triathlon.org/olympics/history). In order to obtain the results efficiently, in a standardized fashion and with minimized human error, a custom python script was written and used to download the desired events. The standard distance for those events was 1.5km/0.93 mile swim, 40 km/24.9 mile cycle, and 10 km/6.2 mile run. Events, where the standard distance was altered due to technical or environmental issues (i.e. reduced course; excluded swim split), were excluded from the analysis.”.
That is right, but still it is not clear if official sprint races included in WTS were analyzed or not. They did not were altered due to technical or environmental issues.
Response: We´ve rewritten, as following: “All data were sourced from the official, publicly available results of the World Triathlon Series (WTS) events from 1989 to 2019 (www.triathlon.org/results) and triathlon races held during the Olympic Games (2004 to 2016) (www.triathlon.org/olympics/history). In order to obtain the results efficiently, in a standardized fashion and with minimized human error, a custom python script was written and used to download the desired events. The standard distance for those events was 1.5km/0.93 mile swim, 40 km/24.9 mile cycle, and 10 km/6.2 mile run. Events, where the standard distance was altered due to technical or environmental issues (i.e. reduced course; excluded swim split), were excluded from the analysis. Of note, sprint-distance races were not included.
“Response: Dear reviewer, our current goal with this manuscript was to establish the cut-off points for Olympic-distance triathlon. Such analysis required a good and representative sample size. We believe that Sprint-distance triathlon needs further development and more events before such analysis can make an impact. Nevertheless, we appreciate the suggestion, and we will consider this analysis in our future research plans. Thank you for this insight!”
I agrre, but If this races were excluded, please include a table(or lines) indicating year by year the number of races analyced per year and participants, male and female. It will give more information to the reader and will help to understand better the paper and data.
Please consider to organize data is I suggest as follows.
Response: Done.
RESULTS
“Please, reconsider organize your data (table 1), not by decades. At least 1989-.midle 90’s (no drafting period), middle 90’s – 2008 (World Championship for a single race) 2009-2019 (WTS). This should be indicated in methods section.
Triathlon races was “different” in these 3 period of time.
Response: Done.
DISCUSSION
Please, consider include discuss results and cut off time divided by those 3 periods of time.
Response: Done.
This paper should be also included in discussion section.
Ofoghi B, Zeleznikow J, Macmahon C, Rehula J, Dwyer DB. Performance analysis and prediction in triathlon. J Sports Sci. 2016;34(7):607-12. doi: 10.1080/02640414.2015.1065341. Epub 2015 Jul 16. PMID: 26177783.
Response: Done.